# Human Uterus Transplantation from Living and Deceased Donors: The Interim Results of the First 10 Cases of the Czech Trial

**DOI:** 10.3390/jcm10040586

**Published:** 2021-02-04

**Authors:** Jiri Fronek, Jakub Kristek, Jaroslav Chlupac, Libor Janousek, Michael Olausson

**Affiliations:** 1Department of Transplantation Surgery, Institute for Clinical and Experimental Medicine, Videnska 1958/9, 140 21 Prague, Czech Republic; jakub.kristek@ikem.cz (J.K.); jaroslav.chlupac@ikem.cz (J.C.); libor.janousek@ikem.cz (L.J.); 2Department of Anatomy, Second Faculty of Medicine, Charles University, V Uvalu 84, 150 06 Prague, Czech Republic; 3First Faculty of Medicine, Charles University, Katerinska 1660/32, 121 08 Prague, Czech Republic; 4Department of Transplantation Surgery, Institute of Clinical Sciences, Sahlgrenska Academy, University of Gothenburg, Sahlgrenska University Hospital, Bla Straket 5, 413 45 Gothenburg, Sweden; michael.olausson@surgery.gu.se

**Keywords:** uterus transplantation, absolute uterine-factor infertility, living donors, deceased donors

## Abstract

Introduction: Uterus transplantation (UTx) is a rapidly evolving treatment of uterine-factor infertility. We report the results of the first 10 UTx procedures performed at our institution. Methods: The program started in April 2016 as a two-arm study comparing the efficacy of UTx from live donors (LD) and deceased donors (DD). Results: Between April 2016 and April 2018, we performed five DD UTx and five LD UTx. Two grafts had to be removed early due to thrombosis. One graft was removed due to chronic rejection and previous herpes simplex infection at month 7. Graft survival is 70% at one year. Recipient survival is 100% at two years. Live donor survival is 100% at three years. Three live-births have been achieved, two from a LD and one from a graft from a nulliparous DD. Vaginal anastomotic stenosis occurred in 63% (5/8) of grafts. Self-expanding stents have shown preliminary suitability for the treatment of vaginal stenosis. Three recipients developed severe acute rejection. Conclusion: The interim results of our study demonstrate mid-term viability in 70% of grafts. The LD UTx produced two live births and the DD UTx produced one live birth. Nulliparous donors should be considered for donation.

## 1. Introduction

Uterus transplantation (UTx) has evolved from a purely experimental to a single method of treatment for women affected with absolute uterine-factor infertility (AUFI) [1,2,3]. Albeit quite novel, the method soon demonstrated its feasibility [4,5] and has rapidly spread to multiple transplant centers over the world. Despite the procedure becoming more common, many aspects of it remain unclear due to its complexity and overall limited amount of experience. Many medical, technical, and ethical issues need to be clarified. For instance, due to the risk of morbidity along with ethical issues associated with the procurement of a graft from a living donor (LD) [6], it is vitally important to establish the potential of grafts procured from deceased donors (DD). Complications up to grade IVa according to the Clavien-Dindo classification [7], have been encountered in LDs and in recipients [3,8,9,10,11]. Although livebirths have been reported from DD grafts [2,12,13], the overwhelming majority of UTx has relied on grafts from LDs [1,14,15,16,17,18]. The aim of this report is to retrospectively present the interim results of our first 10 cases of UTx that represent the first half of the Czech UTx trial using both LDs and DDs.

## 2. Materials and Methods

### 2.1. Study Approval

Our team obtained approval with the trial from the following institutions: 1) the Ministry of Health, Czech Republic (the trial registration number MZDR 32776/2015), and 2) the Joint Ethics Committee of the Institute for Clinical and Experimental Medicine (IKEM) and Thomayer Hospital (number 2044/15 (NM-15-01)). We registered the trial at Clinical Trials Database (identifier NCT03277430). The study approval was for 20 UTx, 10 from LDs, and 10 from DDs.

The trial was performed at the IKEM hospital, Prague, in collaboration with Prof. Michael Olausson, M.D., Ph.D., from the Department of Obstetrics and Gynecology, Sahlgrenska Academy, University of Gothenburg, Gothenburg, Sweden. Additionally, a team of obstetricians and gynecologists from University Hospital Motol, Prague, collaborated in the first half of the trial. All retrieval and transplant procedures were carried out by a senior transplant surgeon and principal investigator of the study (J.F.).

### 2.2. Study Participants

Requests for participation were received by phone or email. After initial phone screening, candidates were invited for a thorough evaluation. The final decision was made by a multi-disciplinary team. Candidate recipients who had no suitable LD were listed for a DD UTx. The nature of the study, including potential risks, was thoroughly explained to all study participants (LDs, recipients, and recipients’ spouses). Written informed consent with the trial and with the publication of the results of the trial was obtained from all graft recipients and LDs. In the Czech Republic, the principle of presumed consent with regards to multi-organ recovery from DDs applies. Additionally, as of December 2020, there is no special legislation regarding recovery of vascular composite allografts (VCA). However, approval of uterus procurement was still obtained from the DD’s next-of-kin in all cases.

### 2.3. Donor’s Evaluation

Inclusion criteria for a LD comprised the following: female 18–60 years of age, ≤4 childbirths, ≤1 caesarian section, and good general health assessed by the study team. The advantages of a LD are a planned procedure, detailed donor evaluation and selection, and shorter warm and cold ischemia time. On the other hand, it does burden the donor with the surgery, which may also result in complications.

A DD provides a graft with longer vascular pedicles and is free of potential complications that could otherwise arise with a LD. After brain death (DBD), all DDs were evaluated according to the protocol of multi-organ donors. No donors are accepted after circulatory death. Eligibility criteria for a DBD comprised the following: general suitability for multi-organ donation, permission for uterus recovery, institutional review board approval, clinical trial registration, AB0 compatibility, negative cross-match, female 18–60 years of age, no previous hysterectomy, and no previous uterus malignancy. The full criteria of uterine DD are published elsewhere [19]. A full gynecological history was not available in all cases at the time of donation from a DD. Much emphasis was put on findings revealed by ultrasound (US), intraoperative findings during surgical exploration, back-table hysteroscopy, and Pap-smear.

Before a uterine donor candidate can be accepted, she requires a thorough assessement, which includes determining the quality of her uterine vessels. At our institution, computed tomography angiography (CTA) is used for evaluation of arteries, i.e., their course, diameter, degree of atherosclerosis, variant anatomy, etc. According to our preliminary results, CTA was insufficient in the evaluation of uterine venous outflow (uterine and utero-ovarian veins) in three of six uterine LD candidates. Whereas all of the LDs were assessed using CTA, only three of five DDs were evaluated in this manner. Magnetic resonance angiography (MRA) is reported to achieve better results in the assessment of uterine veins [9,20]. However, none of the donors presented was evaluated with MRA during the first half of the study.

### 2.4. Recipient’s Evaluation and Protocol

Inclusion criteria for recipients comprised the following: female 18–40 years of age, absolute uterine factor infertility based on congenital or acquired uterus absence, desire for a child, having a male partner and good general health.

The uterus was transplanted in an orthotopic position in the lesser pelvis with arterial and venous anastomoses to the iliac vessels and vaginal anastomosis to the vault of vagina (Figure 1). Vascular anastomoses were fashioned without the administration of intraoperative heparin. The intended period for the graft in situ and exposure to immunosuppression is 5 years. The ultimate objective is a livebirth of 1–2 children. Completion hysterectomy and withdrawal of immune suppression should be carried out thereafter.

To prevent graft thrombosis, all recipients were treated with low-molecular-weight heparin (LMWH) in a prophylactic dose in the first postoperative month. The recipients received 100 mg of acetylsalicylic acid (ASA) daily during the entire post-transplantation period. The antibiotic/antimycotic/chemotherapeutic prophylaxis comprised ampicillin/sulbactam for 24 h, fluconazole for five days, and trimethoprim/sulfamethoxazole for four months (three times a week). Cytomegalovirus (CMV) prophylaxis follows the recommendation for “surveillance after prophylaxis” [21], i.e., 900 mg of valganciclovir daily for 6 months and a follow-up of CMV viremia regularly. This schema was applied to any CMV-serostatus, i.e., D+/R-, D+/R+, D-/R+, and D-/R-. The induction immunosuppression consisted of 1.5 mg/kg of anti-thymocyte immunoglobuline (ATG) intravenously (i.v.), 2 g of mycophenolate mofetil (MMF) daily, 500 mg i.v. of methylprednisolone and tacrolimus (TAC). Based on a negative cytotoxic cross-match, immunosuppression was initiated prior to UTx. Postoperatively, methylprednisolone is tapered down and, on day 8, exchanged for prednisone (Pr) 20 mg, which is slowly tapered off over one month. TAC is adjusted to a trough level of 10–15 µg/mL in the first six weeks and 5–10 µg/mL in later stages. MMF was tapered down from 2 g to 1.5 g, and then to 1 g over a period of three months. After the three-month MMF withdrawal, the patient was left on TAC monotherapy.

### 2.5. Follow-Up

The follow-up comprises four indispensable components: (1) clinical examination; (2) laboratory examination; (3) assessment of perfusion; and (4) cervical biopsies. The frequency of the visits is also defined per-protocol. The recipients are checked twice a week postoperatively in the first month, once every two weeks postoperatively in months 2–6, and once every three weeks in the further course. The visit involves a laboratory examination, a Doppler transvaginal or transabdominal ultrasound (US), and a clinical examination. The clinical examination consists of an abdominal examination and an inspection of the vagina and cervix to look for discoloration or pathological vaginal discharge. Doppler US is a crucial means of follow-up of the perfusion of the graft. Biopsies are planned for postoperative days 7 and 14, and later for months 1, 2, 3, 4, 6, 7, 8, 10, and 12 post-transplantation. During pregnancy, a per-protocol biopsy is performed at the gestational age of 20 weeks. In case of rejection, therapy is determined based on the severity of rejection [22]. While borderline and mild acute rejection (AR) can be treated with increases in TAC maintenance or small doses of methylprednisolone (MP). Moderate and severe AR may mandate the use of high doses of MP and/or ATG and/or azathioprine (AZA).

## 3. Results

### 3.1. Recipients of UTx

In the course of two years, i.e., between April 2016 and April 2018, 10 recipients underwent UTx, namely five DD UTx and five LD UTx. Nine recipients suffered from Mayer-Rokitansky-Küster-Hauser syndrome (MRKHS) type I, and one recipient (Case 7) suffered from MRKHS type II (including agenesis of the right kidney). The mean age of the recipients was 28 ± 3 years (mean ± standard deviation, SD, range 23–33). All transplantations were blood type compatible. The characteristics of the recipients and corresponding donors is provided in Table 1.

In all but one case, the arterial and venous anastomoses were sutured on the recipient´s external iliac vessels. In Case 1, the internal iliac artery and vein were used for right-sided anastomoses. In all but one case, the arterial anastomosis of the graft´s uterine artery was performed using the donor internal iliac artery patch or segment. In Case 1, the anastomosis was carried out directly with the ends of the uterine arteries. Venous outflow was established with two anastomoses in five cases: either two uterine veins or two ovarian/utero-ovarian veins depending on vein size, length and quality. All 4 venous outflow anastomoses were able to be carried out in the remaining five cases. In Case 8, an adjunct third venous outflow anastomosis was added to the two previous ones during venous thrombectomy on day 5. Detailed characteristics of the vascular anastomoses are shown in Table 2.

Operative details and post-operative complications including open or vaginal revision surgeries are listed in Table 3.

### 3.2. Deceased Donors

Firstly, a transvaginal ultrasound is performed to rule out any pathologies of the uterus, such as tumors, scars, intraluminal adhesions, etc. [23]. A median laparotomy via xiphoid-pubis incision is then performed, as is standard in multi-organ recovery. At our institution, we follow a workflow similar to that of Testa et al. [24], i.e., the abdominal transplant team begins approximately 2 h prior to the arrival of the other teams. If, upon thorough exploration of the abdominal cavity, no contraindication is found, all of the procured abdominal organs and their vascular structures, including the infra-renal aorta and inferior vena cava, undergo standard preparation. The uterus is then thoroughly inspected for its quality to rule out any pathologies, e.g., scars, rigidity, myomas, signs of trauma, adhesions, pathologies of the adnexa, atherosclerosis of the pelvic vessels, etc. [23]. If its condition is deemed acceptable, the uterus and uterine vascular structures are prepared in a manner previously described [23]. Once that is complete, the remaining multi-organ recovery is performed as usual with only one difference: cannulation is performed via the external iliac artery and vein. Once cannulation, perfusion with HTK solution (Custodiol^©^, Dr Franz Kohler Chemie, GmBH, Germany), and excision of all other organs are complete, the uterine graft is finally procured. In this way, there is no prolongation of the cold ischemia time (CIT) of life-saving organs. At the back-table, the graft is inspected for perfusion and vessel quality. A hysteroscopy ex vivo is performed on the bench, and the graft, including the vascular pedicles, is prepared for the transplantation. The mean age of DDs was 40 ± 18 years (range 19–57). The average duration of multi-organ procurement, including uterus, from a DBD was 6 h 46 min ± 65 min (range: 4 h 53 min–7 h 39 min, *n* = 5).

### 3.3. Live Donors

The mean age of all donors was 46 ± 14 years (range 19–59) (Table 1). The mean age of LDs was 51 ± 5 years (47–59). At our institution, six LD procurements were performed but only five uterine grafts were transplanted. In one case involving an altruistic donor (LD6, 30 years, 34 BMI), the graft was retrieved but not transplanted due to very diminutive uterine veins and a number of extremely fragile utero-ovarian veins revealed at the back-table. Additonally, to our surprise, the final histology of the cervix indicated high grade dysplasia despite a negative pre-operative Pap-smear test. LDs were mothers of the recipients in four out of five cases; in Case 4 (LD2), the LD was the recipient’s aunt (mother’s sister) since the mother was excluded from donation due to diabetes (Table 1). All procurements from LDs were open laparotomy procedures via an infra-umbilical incisions. Operative details as well as complications are presented in Table 4.

In two cases (LD3 and LD5), adjunct oophorectomy needed to be performed in premenopausal women, resulting in preterm menopause, which was treated pharmacologically. Intra-operative iatrogenic ureteric laceration occurred in LD5. It was treated with a suture and placement of a temporary ureteric JJ stent. The LD4 suffered from urinary bladder hypotonia, which resulted in urinary retention. It was treated with a suprapubic catheter, which was inserted under local anesthesia. The urinary retention issue resolved three months after donation. The altruistic LD6 suffered from hypesthesia of the dorsolateral aspect of her calf in the early and late postoperative course. This was attributed to the defective application of compression stockings. The neural defect resolved slowly through therapy with B-complex vitamins. The three-year survival rate of our live donors was 100%.

### 3.4. Outcomes

#### 3.4.1. Graft and Recipient Survival

Early graft loss occurred twice and mid-term graft loss occurred once. In Case 2, a hematoma was removed and the right uterine artery was re-anastomosed on day 1. Hysterectomy of the graft was carried out on day 7 for an arterial thrombosis that possibly occurred, according to explant histology, in part due to atherosclerosis. In Case 8, removal of a hematoma and thrombectomy of the left uterine vein were performed on day 5. Additional anastomosis of the right ovarian vein was also carried out. However, the graft had to be removed on day 15 due to complete thrombosis. Histology indicated hemorrhagic graft necrosis. Moreover, immunological examination on day 18 revealed positive T-FACS cross-match and the presence of de novo donor-specific antibodies (DSA) (Luminex^©^ assay) against human leukocyte antigens (HLA) class I (B44, 6700 median fluorescence intensity (MFI)).

In Case 7, herpes simplex virus (HSV) infection of the graft was revealed in a protocol cervix biopsy on day 40. This infection was treated with i.v. and p.o. antiviral drugs. No growth of endometrium was noted on US and menstrual bleeding never started. A graft hysterectomy was performed on day 213 post-transplantation. Histology revealed complete fibrous obliteration of the cervical canal and the uterine cavity due to a possible combination of protracted rejection changes (despite no signs of rejection in previous cervical biopsies), vascular thrombosis, graft atherosclerosis and HSV infection. The 30-day graft survival rate or technical success rate was 80%, while the one-year graft survival rate was 70%. In Case 5, completion hysterectomy was performed after successful delivery on day 1109 post-transplant. Six of the remaining grafts are currently viable.

All recipients are alive to this day. The two-year survival rate for recipients is 100%. To date, recipient Cases 1–4 have survived 4 years, Cases 5–9 have survived 3 years and Case 10 has survived 2.7 years. Details on graft and recipient survival are given in Table 5.

#### 3.4.2. Complications in Recipients

##### Vaginal Stenosis

Seven out of 10 of our recipients were born with vaginal agenesis as part of the MRKHS. Six of them underwent a previous laparoscopic Vecchietti procedure in order to create a neo-vagina, and one (Case 5) underwent a dilation procedure. Three recipients were born with a vagina (Cases 1, 7 and 10). One specific complication of UTx surgery is a stenosis of the vaginal anastomosis. Details on vagina type and the occurrence of post-transplant anastomotic stenosis are listed in Table 1. Stenosis occurred in five out of eight viable grafts (63%). The first occurrence of stenosis, which required therapy, appeared in Cases 1, 5, 6, 7, and 10 on post-transplant days 267, 75, 53, 58, and 22, respectively (Table 3). All of the cases needed repeated dilation and/or stent insertion to treat the stenosis.

In Case 1, the stenosis was first dilated by a gynecologist on day 267 and hemostasis was carried out for bleeding caused by a cervix biopsy. Additionally, the gynecologist twice attempted to cut and resect the vaginal stenosis using a diathermy knife. After the last attempt, the patient developed vesico-vaginal fistula; the gynecologist recommended a graft hysterectomy. The transplant team, however, disagreed with the plan, and on day 272, vesico-vaginal fistula closure was attempted by applying a suture trans-vaginally. Unfortunately, the fistula did not heal on a permanent urinary catheter. Therefore, open repair of the fistula, stenosis excision and a vaginal re-anastomosis were carried out on day 341. This procedure was performed by a transplant surgeon and an urologist. The patient healed without complications, yet the stenosis recurred and a stent was inserted on day 383. A total of two more stent insertions were required after the second surgery and vaginal re-anastomosis, after which there was no longer a need for a stent. In Case 5, repeated dilation and stenting was necessary only for the first 10 months post-transplantation. A scheduled transplant hysterectomy was performed five months after successful delivery. In Case 6, repeated dilation and stenting is still necessary to this day. In Case 7, a stent was also inserted; however, this graft was removed due to chronic rejection and HSV infection. The stenosis healed two months prior to the hysterectomy. In Case 10, repeated stenting contiues to this day. This recipient became pregnant after her fourth attempt and lost the child at week 20. After this miscarriage, the stent was inserted only once. After couple of weeks there is no need for further stenting, the vaginal anastomosis stays wide open.

##### Infections

Uncomplicated urinary tract infection (UTI) was treated with antibiotics in 4 recipients (Cases 1, 3, 6, and 7), and repeatedly in Cases 1 and 3. In Case 1, UTI was associated with vesico-vaginal fistula. Description and time occurrence of infections, leukopenia, rejection episodes and renal impairments are listed in Table 6. All these events necessitated therapy. A detailed description of infectious complications will also be provided in another article.

##### Leukopenia

Leukopenia episodes were treated by adjusting existing, and potentially myelo-toxic, therapy as well as by administering filgrastim. In Cases 3 and 5, examination of a bone marrow aspirate was performed, revealing post medication suppression.

##### Rejections

The human leukocyte antigen (HLA) mismatch between the donors and the recipients, percentage of panel-reactive antibodies, acute rejection episodes and their treatments are presented in Table 6. Only Case 4 was free of AR. When excluding three graft losses (two early: Cases 2 and 8; and one mid-term: Case 7), a total of 14 episodes of AR were treated in 6 remaining recipients. Episodes of AR reappeared in 5 of these. Eleven episodes of AR were mild or moderate (grade 1 or 2) and three were severe (grade 3) [22]. Pre-transplant DSA were negative in all cases. Recipient Case 8 developed de novo DSA that were probably associated with early graft loss due to thrombosis (see Section 3.4.1 Graft and Recipient Survival). Recipient Case 6 developed de novo DSA against HLA class II on day 199 (DQ7, 2500 MFI) that were associated with severe AR.

##### Renal Complications

Chronic kidney disease (CKD) emerged in recipient Case 1 as early as day 27 (serum creatinine 114 μmol/L, estimated glomerular filtration rate (eGFR CKD-EPI) 0.83 mL/s). The patient gradually developed CKD G3A1 caused by nefrotoxic medication including calcineurin. Her renal function is permanently impaired and oscillating. Her maximum serum creatinine was 154 μmol/L and her minimum eGFR was 0.63 mL/s on day 1101. Her pregnancy passed without a significant worsening of CKD.

Recipient Case 4 also developed CKD on day 104, and she has sustained a mild reduction in renal function (CKD G2, maximum serum creatinine of 103 μmol/L and minimum eGFR of 1.06 mL/s). Recipient Cases 5 and 6 suffered from a decline in renal function on days 502 and 68, respectively, necessitating hospitalization and i.v. fluids. This occurred in both cases, however, during an episode of inter-current gastroenteritis and exhibited a prerenal etiology and temporary nature.

#### 3.4.3. Pregnancies and Livebirths

All seven recipients with viable grafts underwent embryo transfers (ET). To date, five have become pregnant, of which two have suffered miscarriages (Cases 3 and 10) and three have given birth (Cases 1, 5, and 9). There were no congenital malformations nor neonatal complications. None of the recipients are currently pregnant. The medication of the recipients before commencing ET and/or pregnancy was as follows: Case 1: TAC, Pr, AZA, ASA, Case 3: TAC, Pr, AZA, ASA, Case 4: TAC, ASA, Case 5: TAC, ASA, Case 6: TAC, Pr, ASA, Case 9: TAC, Pr, AZA, ASA, Case 10: TAC, AZA, ASA. A prolonged-release TAC once daily was used in Case 1 (Envarsus^©^, Chiesi) and TAC twice daily was used in the remaining cases (Prograf^©^, Astellas). Gynecologic data on transplanted uterine grafts, embryo transfers, pregnancies and livebirths are presented in Table 7.

In Case 1 (LD UTx), the ETs were slightly delayed due to a vesico-vaginal fistula which required repair. She underwent seven unsuccessful ETs. Owing to numerous unsuccessful ETs, as well as gaps between the transfers, resulting in prolonged exposure to immune suppression, the recipient developed renal impairment. The eighth ET was successful and she gave birth to a healthy baby girl with a birthweight of 2115 g (10–50 percentile) and an Apgar score of 9-10-10. Apart from gestational diabetes treated with diet and placenta praevia marginalis the pregnancy was uneventful. In Case 3 (UTx from a nulliparous DD), the recipient underwent 11 ETs, three of which resulted in clinical pregnancy; however, all three of these ended in miscarriage for unknown reasons. She also suffered from three episodes of AR, the last of which was severe. In Case 4 (LD UTx), the recipient suffered from no complications at all (except for mild CKD). She underwent 11 ETs but failed to achieve pregnancy. A new ovum pick up will be necessary.

In Case 5 (UTx from a nulliparous DD), the recipient suffered from numerous late complications: vaginal stenosis, three episodes of mild AR, four episodes of leukopenia and *Clostridium difficile* colitis with perforated appendicitis [25]. She got pregnant after the fourth ET. Apart from gestational diabetes mellitus necessitating insulin therapy, the pregnancy was uneventful. At a gestational age of 34 + 6, she gave birth to a healthy baby boy with a birthweight of 2740 g (50–90 percentile) and an Apgar score of 7-9-9. This was the first childbirth from a DBD donor in Europe and the first childbirth from a nulliparous graft worldwide. This event has previously been reported [13]. After the delivery, four cervix biopsies were performed. Despite good immunosuppression levels, there was an ongoing moderate to severe graft rejection. Based on the study protocol, such rejection would have required aggressive anti-rejection treatment. Taking into consideration the newborn, lactation and potential risks to the mother, we did not proceed with such treatment. Because the patient did not wish to have a second child, and due to the ongoing rejection, we did not recommend further embryo transfers and instead planned a graft hysterectomy. The patient had her graft removed five months after the C-section. This completion hysterectomy was uneventful. Immunosuppression was tapered off over the next three months.

In Case 6 (LD UTx), the patient underwent five ETs. Pregnancy has yet to be achieved. She suffered from both moderate and severe episodes of AR. Surgery to remove intrauterine adhesions is planned.

In Case 9 (LD UTx), the patient suffered from one moderate and one severe episode of AR. A third ET led to pregnancy. Pregnancy-associated hypertension was treated pharmacologically. The patient gave birth to a healthy baby girl at a gestational age of 36 + 2, with a birth weight of 2300 g (10–50 percentile) and an Apgar score of 10-10-10. The patient has already undergone 2 ETs with the intention of having a second child.

In Case 10 (DD UTx), the recipient had one episode of moderate AR. She went through six ETs. The second ET included two embryos. The fifth ET resulted in pregnancy, but it was discontinued through septic abortion for unknown reason at week 22.

## 4. Discussion

In this clinical trial, a total of 20 UTx procedures were intended. Ten of the uterine grafts were meant to be retrieved from live donors and another 10 grafts from deceased donors. The first 10 transplants of this study (first half of the study) were performed in women with MRKH syndrome and comprised five UTx from LDs and five UTx from DDs. Three recipients experienced graft loss. Three live births of healthy children were achieved, comprising two from LD UTx and one from nulliparous DD UTx.

The present study is limited by the relatively low number of UTx procedures (*n* = 10). However, we feel it is relevant to publish interim results since UTx is still a novel and rapidly evolving field. Moreover, the number of child births in our study (*n* = 3) is still relatively low. One patient had her graft removed after a successful delivery, and another desires a second child and is currently undergoing embryo implants. The remaining four recipients continue receiving embryo transfers. Regrettably, two of them have experienced miscarriages.

In this trial, there were three incidents of graft loss. In Case 2 (LD UTx) and Case 8 (LD UTx), graft loss was attributable to early vascular thrombosis occurring on day 7 and on day 15, respectively. In Case 8, additional immunological examination revealed a newly positive T-FACS cross-match and the presence of de novo DSA against HLA class I. This may suggest immune rejection involvement in the development of the thrombosis. The mid-term loss of the third graft on day 213 was related to an HSV infection. Our technical success rate of 80% is consistent with that of other studies [3,9,15,17]. The ETs in our cohort were delayed in part due to the rejection episodes. With the exception of one patient (Case 4, LD UTx), six out of seven viable grafts sustained at least one episode of AR and required treatment. However, the incidence of AR events did not seem to have a negative effect on pregnancy outcomes. For instance, recipient Case 9 (LD UTx) gave birth to a healthy child despite having experienced previous episodes of severe AR. The other two recipients who gave birth to a healthy baby (Case 1, LD UTx and Case 5, DD UTx) both suffered from 3 episodes of mild to moderate AR. There were no rejection episodes during pregnancies in our study. Three cases that resolved by steroid therapy have been reported worldwide, namely in Gothenburg, Sweden [1], Cleveland, Ohio [12], and Dallas, Texas [26].

Stricture of vaginal anastomosis was a relatively common and protracted complication. It occurred in five of eight (63%) viable grafts. None of the pre-transplant vagina type in MRKHS, i.e., own vagina, dilation or the L-Vecchietti procedure, seemed to affect the incidence of post-transplant stenosis. In fact, stenosis occurred in all three of these scenarios. However, stenosis occurred in all there recipients with a vagina (Cases 1, 7, and 10), in only one recipient after dilation (Case 5), and in only one out of four recipients who received the L-Vecchiettti procedure (Case 6). The other there recipients who received an L-Vecchiettti neovagina were stenosis-free. The presence of stenosis did not appear to significantly affect conception or pregnancy outcomes. Indeed, all recipients who gave birth had stenosis (Cases 1, 5, and 9). Nevertheless, the number of procedures is too low to be able to draw a definitive conclusion.

In the largest series worldwide the rates of mid-term graft viability, vaginal strictures, graft rejections, miscarriages and livebirths were as follows. In the pioneering Gothenburg (Sweden) cohort by Brannstrom, Olausson et al., nine LD UTx have been accomplished and two graft losses have been experienced, one due to thrombosis on day 3 and another one due to infection on day 105 [27]. During the first postoperative year, 10 episodes of mild rejection were detected in five recipients [28]. From the seven vital grafts, eight healthy babies were born to six women. Thus, the take-home-baby rate was 85% (6/7) in vital grafts. All live births were singletons after single ETs. One patient had repeated miscarriages, some as late as gestational week 15, and did not complete a pregnancy with live birth. Vaginal strictures were not reported [27].

In a subsequent Gothenburg study, 8 robot-assisted uterus retrievals from LDs andopen UTx have been carried out, with mid-term viability in 6/8 cases [29]. The interim outcomes in the 6 vital grafts are as follows: stenosis of vaginal anastomosis 2/6, graft rejection 2/6 (one grade 1 and one grade 2), rate of miscarriages 0/6, and livebirths 1/6 [30].

In the study by Brucker et al. (Tubingen, Germany), four LD UTx have been done with full mid-term graft viability in all of the four recipients. Stenosis of the vaginal anastomosis was not reported and mild graft rejection happened in 2/4 recipients. One early miscarriage at eight weeks and four days of pregnancy was experienced. Two livebirths have been achieved in the first two UTx women, including the one with previous pregnancy loss [5].

In the DUETS (Dallas UtErus Transplant Study) by Testa, Johannesson et al., 20 UTx have been performed, 18 from a LD and two from a DD. Six graft failures occurred. In phase 1, 50% of recipients (5/10) had a technically successful uterus transplant, compared to 90% (9/10) in phase 2 [3]. In the first six months post-transplant, seven episodes of acute cellular rejection requiring treatment with a steroid cycle were diagnosed. One organ rejection episode was detected during pregnancy and was resolved with steroids. No miscarriage was reported. The occurrence of vaginal strictures necessitating dilation was in 8/14 (57%) of the grafts. All of these recipients were born with a vagina. Of the 14 technically successful transplants, 12 live births occurred in 11 patients. Ten recipients delivered one neonate, and one recipient delivered two neonates. Thus far, the live birth rate per attempted transplant was 55%, and the live-birth rate per technically successful transplant was 79% [26].

One challenging aspect of UTx is utilizing nulliparous grafts for transplantation. The main issue is that these grafts have yet to prove their ability to promote childbirth. The first UTx from a DD ever had used a nulliparous DD and this was also the second UTx case worldwide, carried out in Turkey in 2011. This case produced clinical pregnancy but ended in a miscarriage [31,32]. In our study, one nulliparous graft was retrieved from an altruistic LD. This graft, however, was not utilized for transplantation owing to its very thin vasculature and was discarded. Another graft from a nulliparous deceased donor was successfully transplanted (Case 5, DD UTx), and despite numerous complications [25], this recipient gave birth to a healthy child. This was the first livebirth from a DD in Europe and the first ever from a nulliparous graft [13]. Completion hysterectomy was performed thereafter. A second graft from a nulliparous DD was also transplanted (Case 3, DD UTx). This patient underwent 11 ETs, but unfortunately experienced three miscarriages. She also experienced grade 3 AR, which necessitated treatment with MP, AZA and ATG. In summary, using nulliparous grafts for UTx is feasible; however, more experience is needed with this type of graft. 

Recently, more than 70 UTx procedures and 23 livebirths from 21 recipients have been reported worldwide [18]. Uterine grafts from LDs were used in the majority of these cases. To date, only three live births resulting from grafts taken from DDs have been reported, including one of our own [2,12,13]. Though UTx from both LDs and DDs is feasible and can lead to successful childbirth, the efficacy of LDs versus DDs has yet to be confirmed due to the low number of DD UTx cases.

## 5. Conclusions

Ten uterus transplants, comprising five from live donors and five from deceased donors, have been performed at our institution. The interim results of our study demonstrate mid-term viability for 70% of uterine grafts. Stenosis of vaginal anastomosis was encountered in five of eight grafts (63%) and necessitated repeated stent placement. Immunosuppression side effects, infectious complications and recurrent rejection may have led to significant delays in embryo transfers.

Three live births have been achieved, two from a LD and one from a nulliparous DD. Therefore, nulliparous donors can be considered for donation; however, their potential needs further clarification. We also observed four miscarriages in two recipients. This preliminary report gives further evidence of the feasibility of uterine transplantation from both living and deceased donors.

## Figures and Tables

**Figure 1 jcm-10-00586-f001:**
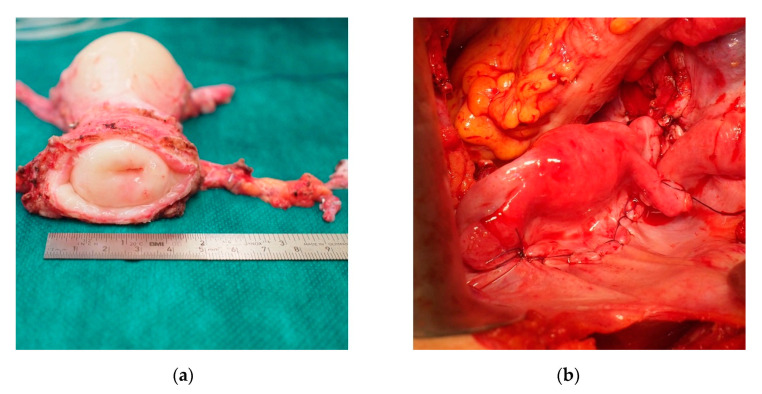
(**a**) Uterus graft from a deceased donor during back-table surgery. (**b**) Uterus graft transplanted in an orthotopic position in the lesser pelvis.

**Table 1 jcm-10-00586-t001:** Demographics of 10 consecutive recipients and donors of uterus transplantations. Both donor and recipient surgeries were open laparotomy procedures. The LD2, aunt was the recipients´s mother´s sister.

Recipients							Donors			
Case	Age	BMI	Bl. t.	Indic.	Vagina Type	Vaginal An. Stenosis	Donor Type	Age	BMI	Bl.t.
1	30	22	AB+	MRKH	Own vagina	Yes	LD1 (mother)	53	30	AB+
2	29	20	0+	MRKH	L-Vecchietti	N/A (graft loss day 7)	DD1	57	20	0+
3	26	20	A+	MRKH	L-Vecchietti	No	DD2	24	33	0-
4	26	18	A+	MRKH	L-Vecchietti	No	LD2 (aunt)	58	34	A+
5	24	21	AB+	MRKH	Dilatation	Yes	DD3	19	24	0+
6	23	26	B-	MRKH	L-Vecchietti	Yes	LD3 (mother)	47	36	0-
7	32	17	A+	MRKH *	Own vagina	Yes	DD4	56	22	A+
8	25	19	0+	MRKH	L-Vecchietti	N/A (graft loss day 15)	LD4 (mother)	49	19	0+
9	26	21	A+	MRKH	L-Vecchietti	No	LD5 (mother)	48	22	A+
10	29	31	A+	MRKH	Own vagina	Yes	DD5	45	32	0+
M ± SD	28 ± 3	22 ± 4				63% (5/8)		46 ± 14	27 ± 6	

* right kidney agenesis. Abbreviations: age (years), anastomosis (an.), blood type (Bl.t.), body mass index (BMI, kg/m^2^), deceased donor (DD), indication (Indic.), laparoscopic (L), live donor (LD), mean (M.), Mayer–Rokitansky–Kűster-Hauser syndrome (MRKH), mean ± standard deviation (SD) and not applicable (N/A).

**Table 2 jcm-10-00586-t002:** Vascular anastomosis details in 10 consecutive uterus transplant recipients.

		Arterial Anastomosis				Venous Anastomosis			
		Right		Left		Right		Left	
Case	Donor	Graft	Recip.	Graft	Recip.	Graft	Recipient	Graft	Recipient
1	LD1	UA end	IIA	UA end	EIA	UOV	IIV	UOV	EIV
2	DD1	UA iliac patch	EIA	UA iliac patch	EIA	UV	EIV	UV	EIV
						UOV	EIV	UOV	EIV
3	DD2	UA iliac segment	EIA	UA iliac segment	EIA	UV	EIV	UV	EIV
4	LD2	UA iliac segment (a.d.)	EIA	UA iliac patch	EIA	UOV	EIV	UOV	EIV
5	DD3	UA iliac segment	EIA	UA iliac segment	EIA	UV	EIV	UV	EIV
						UOV	EIV	UOV	EIV
6	LD3	UA iliac segment (a.d.)	EIA	UA iliac segment (a.d.)	EIA	OV	EIV	OV	EIV
7	DD4	UA iliac segment	EIA	UA iliac segment	EIA	UV	EIV	UV	EIV
						UOV	EIV	UOV	EIV
8	LD4	UA patch	EIA	UA patch	EIA	UV	EIV	UV	EIV
						UOV *	EIV		
9	LD5	UA iliac segment (a.d.)	EIA	UA iliac segment (a.d.)	EIA	UV	EIV	UV	EIV
						OV	EIV	OV	EIV
10	DD5	UA iliac segment	EIA	UA iliac segment	EIA	UV	EIV	UV	EIV
						UOV	EIV	UOV	EIV

* during revision on day 5. Abbreviations: anterior division of IAA (a.d.), deceased donor (DD), external iliac artery (EIA), external iliac vein (EIV), internal iliac artery (IIA), internal iliac vein (IIV), live donor (LD), ovarian vein (OV), recipient (Recip.), uterine artery (UA), uterine vein (UV), and utero-ovarian vein (UOV).

**Table 3 jcm-10-00586-t003:** Operative details and post-operative complications of 10 consecutive uterus transplant recipients. All surgeries were open laparotomy procedures. There were no intra-operative complications. Only the first episode necessitating therapy of a respective type of complication is mentioned.

Recipient Case	Time (Min)	EBL (mL)	PRBC (Units)	MT (Min)	CIT (Min)	LOS (Days)	Post-Operative Complications Clavien Grade
1	209	1200	3	142	383	14	II recurrent UTI, ACR gr 1 on day 459, CMV replication on day 1172, CKD on day 27 IIIb: dilation of stenosis on day 267, vaginal suture for vesico-vaginal fistula on day 272, open repair of vesicovaginal fistula and vaginal reanastomosis on day 341, vaginal stenosis (stent) on day 383
2	245	200	15	100	394	19	IIIb removal of hematoma, reanastomosis of right UA on day 1, hysterectomy for thrombosis on day 7
3	250	1000	0	63	146	9	II: recurrent UTI, AR gr 1 on day 20, leukopenia on day 40
4	229	600	0	84	283	11	
5	299	500	1	91	549	8	II: AR gr 1 on day 13, leukopenia on day 89, IIIb: vaginal stenosis, dilation on day 75, *Clostridium difficile* colitis and acute appendicectomy on day 502
6	296	700	0	119	304	14	II: UTI, AR gr 2 on day 150, leukopenia on day 88, CMV replication on day 374, EBV replication on day 1017 IIIb: vaginal stenosis (stent) on day 53
7	233	300	0	63	246	7	II: UTI, leukopenia on day 72, graft HSV infection on day 40 IIIb: vaginal stenosis (dilation) on day 58, hysterectomy for HSV infection and protracted rejection on day 213
8	216	200	7	44	234	17	IIIb: venous thrombectomy on day 5, hysterectomy for venus thrombosis on day 15
9	216	600	5	64	288	10	II: AR gr 2–3 on day 395 IIIb: hemostasis on day 2
10	319	200	2	81	307	10	II: leukopenia on day 50, AR gr 2 on day 105, CMV replication on day 547 IIIb vaginal hemostasis post biopsy on day 14, vaginal stenosis (stent) on day 22
mean ± SD	251 ± 40	550 ± 347	3 ± 5	85 ± 29	313 ± 109	12 ± 4	

Abbreviations: acute rejection (AR), chronic kidney disease (CKD), cold ischemic time (CIT, min), cytomegalovirus (CMV), Epstein-Barr virus (EBV), estimated blood loss (EBL, mL), grade (gr.), graft manipulating time (MT, min), herpes simplex virus (HSV), length of hospital stay (LOS, days), packed red blood cells (PRBC) given during the course of the entire hospitalization, standard deviation (SD), urinary tract infection (UTI), and uterine artery (UA).

**Table 4 jcm-10-00586-t004:** Operative details and post-operative complications of live donors of uterine grafts. All surgeries were open laparotomy procedures.

Live Donor	Time (min)	EBL (mL)	PRBC (Units)	LOS (Days)	OE	Survival (Days)	Complications Clavien Grade
LD1	321	100	0	7		1676	
LD2	369	800	0	7		1473	
LD3	431	100	0	7	yes	1374	II: preterm menopause treated with HRT
LD4	326	100	0	11		1153	IIIa: urinary bladder hypotonia, day 3, suprapubic drainage under local anesthesia, resolved after 3 m
LD5	332	1000	5	9	yes	1107	II: preterm menopause treated with HRT IIIa: intraoperative ureteric laceration, primary repair and ureteral stent
(LD6)	341	100	0	7		1106	II: lateral sural cutaneous nerve paresis, temporary Graft not used for thin vasculature.
mean ± SD	353 ± 42	367 ± 418	1 ± 2	8 ± 2		3y 100%	

Abbreviations: estimated blood loss (EBL, mL), hormone replacement therapy (HRT), length of hospital stay (LOS, days), months (m), oophorectomy (OE), packed red blood cells (PRBC) given during the course of the entire hospitalization, standard deviation (SD) and years (y).

**Table 5 jcm-10-00586-t005:** Outcomes of 10 consecutive uterus transplantation grafts and recipients.

Case	Donor	Graft Survival (Days)	Recipient Survival (Days)
1	LD1	Viable, 1695 days	1695
2	DD1	Graft loss, thrombosis, day 7	1655
3	DD2	Viable, 1572 days	1572
4	LD2	Viable, 1492 days	1492
5	DD3	Planned hysterectomy, day 1109	1429
6	LD3	Viable, 1393 days	1393
7	DD4	Graft loss, HSV infection, day 213	1338
8	LD4	Graft loss, thrombosis, day 15	1172
9	LD5	Viable, 1126 days	1126
10	DD5	Viable, 984 days	984
		30 days: 80%, 1 year: 70%	2 years: 100%

Abbreviations: deceased donor (DD), Herpes simplex virus (HSV) and live donor (LD).

**Table 6 jcm-10-00586-t006:** Complications of 10 consecutive uterus transplant recipients regarding infection, leukopenia, rejection and renal impairment.

Case	Donor	CMV D/R	Infection	Leukopenia	HLA m/m	PRA	Rejection	CKD
1	LD1	+/+	Recurrent UTI, CMV replication day 1172		2/1	0%	Gr 1, day 459, MP Gr 1–2, day 1053, MP, ATG, AZA Gr 1, day 1270, TAC elev.	G3A, day 27
2	DD1	+/+	N/A, graft loss day 7		4/2	0%		
3	DD2	+/+	Recurrent UTI, *Ureaplasma* 1157	Day 40, 156, 189 (aBM), 1066	4/1	3%	Gr 1, day 19, MP Gr 1, day 372, MP, AZA Gr 2–3, day 930, MP, AZA, ATG	
4	LD2	+/-			1/1	0%		G2, d. 104
5	DD3	+/+	*Cl.difficie* day 502	Day 89, 95, 109, 115 (aBM), 473, 523	3/1	0%	Gr 1, day 13, MP Gr 1, day 193, MP Gr 1, day 367, AZA	Prerenal day 502
6	LD3	+/-	UTI, CMV replication day 374, EBV replication day 1017	Day 88	1/1	0%	Gr 2, day 150, MP Gr 3, day 197, ATG	Prerenal day 68
7	DD4	+/+	UTI, graft HSV infection day 40, graft loss day 213	Day 72	3/2	7%	Protracted rejection in explant	
8	LD4	+/+	N/A, graft loss day 15		2/1	10%		
9	LD5	+/+			1/1	0%	Gr 2–3, day 395, MP, AZA Gr 1–2, day 415, MP	
10	DD5	+/+	CMV replication on day 547	Day 50	2/2	7%	Gr 2, day 105, MP, AZA	
				Mean ± SD	2.3/1.3			

Abbreviations: antithymocyte globuline (ATG), aspiration of bone marrow (aBM), azathioprine (AZA), cytomegalovirus (CMV), day (d.), decesased donor (DD), Epstein–Barr virus (EBV), grade (gr, G), herpes simplex virus (HSV), human leukocyte antigen (HLA) class I/II mismatch (HLA m/m), live donor (LD), methylprednisolone (MP), not applicable (N/A), pre-transplant panel-reactive antibodies (PRA), standard deviation (SD), tacrolimus (TAC) and urinary tract infection (UTI).

**Table 7 jcm-10-00586-t007:** Pregnancy outcomes of 10 consecutive uterus transplantations.

Case	Donor	Grav.	Par.	Menop.	Menses (Day)	Embryo Transfers (Day Post-Transplant)	Miscarriage Days to ET/Gest. Day	Live Birth (Day)
1	LD1	2	2	yes	45	8× (551, 641, 711, 858, 947, 977, 1031, 1482→P)		1713, girl, 35 w + 3
2	DD1	N/R	1	yes	N/A	N/A Graft loss, day 7	-	-
3	DD2	0	0	no	103	11× (245, 322, 405, 537, 638→M, 908, 1204, 1234, 1268, 1368→M, 1533→M)	3× (638/48, 1368/42, 1533/49)	
4	LD2	2	2	yes	133	11× (340, 417, 459, 578, 648, 814, 921, 963, 1095, 1192, 1322)		
5	DD3	0	0	no	58	4× (414, 443, 627, 725→LB)		950, boy, 34 w + 6
6	LD3	2	1s.c.	yes	35	5× (463, 582, 676, 1200, 1236)		
7	DD4	N/R	1	yes	no	N/A Graft loss, day 213	-	-
8	LD4	4	3	no	N/A	N/A Graft loss, day 15	-	-
9	LD5	2	2	no	86	5× (282, 373, 490→LB, 1079, 1114)		721, girl, 36 w + 2
10	DD5	N/R	N/R	no	26	6× (433, 469 (2 emryos), 503, 645, 719→M, 964)	1×, 719/137	
			Mean ± SD		69 ± 39			

Abbreviations: caesarian section (s.c.), deceased donor (DD), embryo transfer (ET), gestational (gest.), gravidity (Grav.), live birth (LD), live donor (LD), menopause (Menop.), miscarriage (M), not applicable (N/A), not reported (N/R), parity (Par.), pregnancy (P), standard deviation (SD) and week (w).

## Data Availability

The data presented in this study are available on request from the corresponding author. The data are not publicly available due to confidentiality.

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
