# Peer review of "Human Uterus Transplantation from Living and Deceased Donors: The Interim Results of the First 10 Cases of the Czech Trial"

_jcm, 2021, doi:10.3390/jcm10040586_

Round 1

Reviewer 1 Report

In this article, Dr Fronek and colleagues provided data from their study on Uterus transplantation (UTx). In a two-arm study the authors compared UTx from live 18 donors (LD) and deceased donors (DD). The study 26 demonstrates 70% graft viability and successful pregnancy in two women receiving UTX from DD and LD, respectively. Since UTX is getting more attention currently, this paper falls within the scope of the JCM. The structure of the article makes it easy to read and follow. However, this Reviewer has some methodological concerns and the literature review seem to be not up-to-date.

Overall, the article makes an important contribution to the field by drawing our attention to this clinically relevant topic and pointing out the lack of data.  

Specific comments:

  1. It would be more useful to start with introducing data from recent studies on UTX such as PMID: 32756295, PMID: 30611405.
  2.  It would be easier for the readers if the authors could provide a supplementary table of Inclusion criteria for DD. Did the donors or their relatives specifically agree to the donation of the uterus?
  3. Figure 1 (a) Please provide a scale for the picture. Is this an uterus after a DD or LD?
  4. Could the authors provide some information on their intra- and postoperative anticoagulation protocols?  
  5. One of the most common causes of the allograft rejection is immunization against HLA- class I and II antigens. It is not clear which assays were performed to screen for HLA incompatibility in this program. What were the clinical consequence of anti-HLA antibodies? Please discuss these issues in more details.   
  6. Discussion: One recipient of another LD graft is currently over 30 weeks pregnant. Please give an update on this pregnancy.
  7. Please discuss potential reasons for the miscarriages.  
  8. The authors should address differences in mid-term viability, stenosis of vaginal anastomosis, graft rejection, rate of miscarriages and livebirths between the current study and previous observations. 

Author Response

Dear editors and reviewers,

Thank you very much for your valuable comments and recommendations for the major revision of our manuscript jcm-1079572. Thank you also for giving us the extended time for this revision. We have addressed all your comments and all your suggestions as follows.

Additionally, English language revision by a native speaker has been carried out even this was not explicitly requested. However, we feel that this has been beneficial for the quality of the article.

We enclose a thoroughly revised version of our manuscript. All changes to the text are highlighted in MS Word track changes mode. These include both (1.) changes according to the reviewer´s specific comments and (2.) English edit. The reviewer´s specific comments are addressed in detail below. The English edits are not specifically described but they are visible in the tracking change mode.

For your convenience we also enclose the same manuscript in which all the changes have been done and accepted but are not highlighted (MS Word file: jcm-1079572-major revision accepted changes).

Sincerely

Jiri Fronek

Response to reviewers: jcm-1079572

Reviewer 1.

Specific comments:

  1. It would be more useful to start with introducing data from recent studies on UTX such as PMID: 32756295, PMID: 30611405.

Author: Citations of these recent studies have been added to the Introduction section, line 36, citations 4 and 5.

  1. It would be easier for the readers if the authors could provide a supplementary table of Inclusion criteria for DD. Did the donors or their relatives specifically agree to the donation of the uterus?

Author: The sentence: “Eligibility criteria for a DBD comprised: female 18-60 years of age, no previous hysterectomy, no previous uterus malignancy.” has been extended: “Eligibility criteria for a DBD comprised the following: general suitability for multi-organ donation, permission for uterus recovery, institutional review board approval, clinical trial registration, AB0 compatibility, negative cross-match, female 18-60 years of age, no previous hysterectomy and no previous uterus malignancy.” Line 79-82.

We feel that a supplementary table with detailed DBD criteria should not be included for following reasons: it has been previously published by the authors and the table is rather extensive (Table 2 in citation 19, Kristek, J., L. Johannesson, G. Testa, R. Chmel, M. Olausson, N. Kvarnstrom, N. Karydis and J. Fronek (2019). "Limited Availability of Deceased Uterus Donors: A Transatlantic Perspective." Transplantation 103(12): 2449-2452.)   

In the Czech Republic, presumed consent for organ and tissue retrieval from deceased donors is applied. Despite this, yes, at the time of donation, the consent was obtained from relatives pertaining the retrieval of the uterus graft, because this type of retrieval is novel. This information is given in the manuscript, lines 66-70.

  1. Figure 1 (a) Please provide a scale for the picture. Is this an uterus after a DD or LD?

Author: There is a metallic scale in the picture 1(a) used during sterile surgery procedures.

Uterus graft “from a deceased donor” … was added in the figure legend. Line 107. 

  1. Could the authors provide some information on their intra- and postoperative anticoagulation protocols?

Author: For the information on intraoperative anticoagulation protocol, a new statement was added: “Vascular anastomoses were fashioned without the administration of intraoperative heparin“ Lines 100-101.

In our opinion, the postoperative anticoagulation protocol in sufficiently described in lines 110-112: “To prevent graft thrombosis, all recipients were treated with low-molecular-weight heparin (LMWH) in a prophylactic dose in the first postoperative month. The recipients received 100 mg of acetylsalicylic acid once daily in during the entire post-transplantation period. “.   

  1. One of the most common causes of the allograft rejection is immunization against HLA- class I and II antigens. It is not clear which assays were performed to screen for HLA incompatibility in this program. What were the clinical consequence of anti-HLA antibodies? Please discuss these issues in more details.

Author: Specification was added (Luminex© assay) in line 234.

In Case 8 anti-HLA antibodies were found by Luminex assay on day 18, i.e. 3 days after graft loss for arterial thrombosis on day 15. This is described in Lines 232-235. In Case 6 anti-HLA antibodies were part of diagnosis of severe rejection (Line 302-303) that was treated with ATG, please see Table 6, Case 6 in Line 285.

We believe that clinical consequences of anti-HLA antibodies are already sufficiently reported in the section 3.3.2 Complications in recipients, paragraph: Rejection: “Pre-transplant DSA were negative in all cases. Recipient Case 8 developed de novo DSA that were probably associated with early graft loss due to thrombosis (see above section 3.3.1 Graft and recipient survival). Recipient Case 6 developed de novo DSA against HLA class II on day 199 (DQ7, 2500 MFI) that were associated with severe AR.” Lines 301-303.

  1. Discussion: One recipient of another LD graft is currently over 30 weeks pregnant. Please give an update on this pregnancy.

Author: An update on this pregnancy that resulted in the third live birth has been given throughout the whole manuscript. For example:

Line 23: “Three live-births have been achieved..”

Line 27: “The LD UTx produced two live births and DD UTx produced one live birth.”

Line 308: “Her pregnancy passed without significant worsening of CKD.“

Line 320-21: „… three have given birth (Cases 1, 5 and 9). None of the recipients are currently pregnant.“

Line 323: Table 7 has been updated accordingly

Line 365: Deleted: “Two live births of a healthy child have been achieved, first from UTx from a LD and second from UTx from a nulliparous DD. One recipient of another LD graft is currently over 30 weeks pregnant.“ New sentence: „Three live births of a healthy children were achieved have been achieved, comprising two from LD UTx from a LD and one from nulliparous DD UTx.“

Line 370: Removed: „However, another recipient is > 30 weeks pregnant.“

Line 330: “The eighth ET was successful and she gave birth to a healthy baby girl with a birthweight of 2115 g (10.-50. percentile) and an Apgar score of 9-10-10. Apart from gestational diabetes treated with diet and placenta praevia marginalis the pregnancy was uneventful.”

Line 379-380: Adapted sentence: „The other two recipients who gave birth to a healthy baby (Case 1, LD UTx and Case 5, DD UTx) both suffered from 3 episodes of mild to moderate AR.“

Line 368: Deleted: “Recipient Case 5 who produces a childbirth had a stenosis and recipient Case 9 who also produced a childbirth had no stenosis. Recipient Case 1 who is now > 30 weeks pregnant also had a stenosis.” And replaced by: “Indeed, all recipients who gave birth had a stenosis (Cases 1, 5 and 9)”.

Line 422: Deleted: “Two livebirths have been achieved, one from a LD and one from a nulliparous DD.” New statement: “Three livebirths have been achieved, two from LD and one from a nulliparous DD.”

Line 424: Deleted: “Another recipient of a LD graft is currently over 30 weeks of her pregnancy.” 

  1. Please discuss potential reasons for the miscarriages.  

Author: In Case 3 we do not know the reason for the 3 miscarriages. The sentence in the manuscript was adapted: “… all 3 of themse terminated asended in miscarriage for unknown reasons..“ Line 334.

In Case 10 the reason was infection but of unknown source. We adapted the sentence: “… discontinued through septic abortion for unknown reason at week 22.” - Line 358.

  1. The authors should address differences in mid-term viability, stenosis of vaginal anastomosis, graft rejection, rate of miscarriages and livebirths between the current study and previous observations. 

Author:  These information were added to the discussion, lines 419-441:

“In the largest series worldwide the rates of mid-term graft viability, vaginal strictures, graft rejections, miscarriages and livebirths were as follows. In the pioneering Gothenburg (Sweden) cohort by Brannstrom, Olausson et al., 9 LD UTx have been accomplished a 2 graft losses have been experienced, one for thrombosis on day 3 and another one for infec-tion on day 105 [27]. During the first postoperative year, 10 episodes of mild rejection were detected in 5 recipients [28]. From the 7 vital grafts, 8 healthy babies were born in 6 women. Thus, the take-home-baby rate was 85% (6/7) in vital grafts. All live births were singletons after single ETs. One patient had repeated miscarriages, some as late as gestational week 15, and did not complete a pregnancy with live birth. Vaginal strictures were not reported [27].

In a subsequent Gothenburg study, 8 robot-assisted LD UTx have been carried out, with mid-term viability in 6/8 cases [29]. The interim outcomes in the 6 vital grafts are as follows: stenosis of vaginal anastomosis 2/6, graft rejection 2/6 (one grade 1 and one grade 2), rate of miscarriages 0/6, and livebirths 1/6 [30].

In the study by Brucker et al. (Tubingen, Germany), 4 LD UTx have been done with full mid-term graft viability in all of the 4 recipients. Stenosis of the vaginal anastomosis was not reported and mild graft rejection happened in 2/4 recipients. One early miscarriage at 8 weeks and 4 days of pregnancy was experienced. Two livebirths have been achieved in the first 2 UTx women, including the one with previous pregnancy loss [5].

In the DUETS (Dallas UtErus Transplant Study) by Testa, Johannesson et al., 20 UTx have been performed, 18 from a LD and 2 from a DD. Six graft failures occurred. In phase 1, 50% of recipients (5/10) had a technically successful uterus transplant, compared to 90% (9/10) in phase 2 [3]. In the first 6 months post-transplant, 7 episodes of acute cellular rejection requiring treatment with a steroid cycle were diagnosed. One organ rejection episode was detected during pregnancy and was resolved with steroids. No miscarriage was reported. The occurrence of vaginal strictures necessitating dilation was in 8/14 (57%) of the grafts. All of these recipients were born with a vagina. Of the 14 technically successful transplants, 12 live births occurred in 11 patients. Ten recipients delivered one neonate, and one recipient delivered two neonates. Thus far, the live birth rate per attempted transplant was 55%, and the live-birth rate per technically successful transplant was 79% [26].”

Reviewer 2 Report

General impression

This is an interim analysis of a study about uterus transplantation. The data are of paramount significance in this field, as there <100 published cases of uterus transplantation until today, and there is urgent need for good quality data for the optimization of the protocols.

The submission holds a personal non-scientific tone at some points.

Advantages:

  • Well-written, the results are presented in a very understandable format
  • The results add nicely to the already published literature

Disadvantages:

  • Small numbers
  • Statistics cannot be drawn for comparison between Live donors & Donors after Braing Death regarding the post-transplantation and the pregnancy parameters
  • Retrospective study

Abstract

  1. Page 1, Line 26. ‘Three recipients developed severe acute rejection.’ There were 14 episodes of AR in 6 recipients (Lines 274-276). In case of inconsistence with the data, please remove this sentence.

Introduction

  1. Page 1, Line 26. ‘Two live births have been 46 achieved so far.’ This is not part of the Aim of the study. Please remove the sentence.

Results

  1. Page 7, Line 186. ‘In fact, we were 186 lucky not transplanting this graft for vascular reasons.’ This is not part of the Results. Please remove the sentence.
  2. Page 7, Line 217. ‘3.3. Outcomes’. What about the restoration of menstruation in the recipients: Frequency, Pain, Bleeding, is there any menstrual diary available? What about sexual life: is there any questionnaire available, eg FSFI?
  3. Page 8, Line 243.’ …vaginal suture was applied for confirmed vesico-vaginal…’. Do the authors mean attempt of transvaginal vesico-vaginal fistula restoration?
  4. Page 10, Line 217. ‘3.3.3. Outcomes’. Please provide any adaptations of the medical treatment during pregnancy and the medication the patients initiate each time they get pregnant.
  5. Page 10, Table 7. Any information about placentation? Any biopsies obtained from placentas post-delivery regarding evidence of rejection? Neonatal status: Percentile of Birthweight, Apgar & Congenital malformations.
  6. Page 10, Table 303. What is exactly the pathophysiologic association between embryo-transfer and renal impairment. Please provide reference.

Author Response

Prague, Czech Republic, 26 Jan 2021

Dear editors and reviewers,

Thank you very much for your valuable comments and recommendations for the major revision of our manuscript jcm-1079572. Thank you also for giving us the extended time for this revision. We have addressed all your comments and all your suggestions as follows.

Additionally, English language revision by a native speaker has been carried out even this was not explicitly requested. However, we feel that this has been beneficial for the quality of the article.

We enclose a thoroughly revised version of our manuscript. All changes to the text are highlighted in MS Word track changes mode. These include both (1.) changes according to the reviewer´s specific comments and (2.) English edit. The reviewer´s specific comments are addressed in detail below. The English edits are not specifically described but they are visible in the tracking change mode.

For your convenience we also enclose the same manuscript in which all the changes have been done and accepted but are not highlighted (MS Word file: jcm-1079572-major revision accepted changes).

Sincerely

Jiri Fronek

Response to reviewers: jcm-1079572

Reviewer 2

Abstract

  1. Page 1, Line 26. ‘Three recipients developed severe acute rejection.’ There were 14 episodes of AR in 6 recipients (Lines 274-276). In case of inconsistence with the data, please remove this sentence.

Author: We believe there is no need to remove this sentence. There is consistency in the data. The fact is that there were three episodes of severe acute rejection (grade 3). This fact is given both in the Abstract: “Three recipients developed severe acute rejection.“ Line 26 and in the Results: „… a total of 14 episodes of AR were treated in 6 remaining recipients. The episodes of AR reappeared in 5 of these. Eleven episodes of AR were mild or moderate (grade 1 or 2) and three episodes were severe (grade 3).” Lines 298-300.

Introduction

  1. Page 1, Line 26. ‘Two live births have been achieved so far.’ This is not part of the Aim of the study. Please remove the sentence.

Author: In the jcm-template for MS Word there is a recommendation for the Introduction:  “Finally, briefly mention the main aim of the work and highlight the principal conclusions.” However, the sentence has been removed, line 47.

Results

  1. Page 7, Line 186. ‘In fact, we were lucky not transplanting this graft for vascular reasons.’ This is not part of the Results. Please remove the sentence.

Author: This sentence has been deleted. Line 203.

  1. Page 7, Line 217. ‘3.3. Outcomes’. What about the restoration of menstruation in the recipients: Frequency, Pain, Bleeding, is there any menstrual diary available? What about sexual life: is there any questionnaire available, eg FSFI?

Author: The onset of menstruation post-transplantation is reported in Table 7 for all recipients, Line 324. Unfortunately there are no menstrual calendars available nor data on sexual life after UTx.

  1. Page 8, Line 243.’ …vaginal suture was applied for confirmed vesico-vaginal…’. Do the authors mean attempt of transvaginal vesico-vaginal fistula restoration?

Author: Yes, this was meant. To make the statement more clear, the sentence: “… vaginal suture was applied for confirmed vesico-vaginal fistula.“ was replaced by: „… vesico-vaginal fistula closure was attempted by applying a suture trans-vaginally.“ Line 264.

  1. Page 10, Line 217. ‘3.3.3. Outcomes’. Please provide any adaptations of the medical treatment during pregnancy and the medication the patients initiate each time they get pregnant.

Author:  New sentence was added: “The medication of the recipients before commencing ET and/or pregnancy was as follows: Case 1: TAC, Pr, AZA, ASA, Case 3: TAC, Pr, AZA, ASA, Case 4: TAC, ASA, Case 5: TAC, ASA, Case 6: TAC, Pr, ASA, Case 9: TAC, Pr, AZA, ASA, Case 10: TAC, AZA, ASA. A prolonged-release TAC once daily was used in Case 1 (Envarsus©, Chiesi) and TAC twice daily was used in the remaining cases (Prograf©, Astellas). ” Line 322-325. 

  1. Page 10, Table 7. Any information about placentation? Any biopsies obtained from placentas post-delivery regarding evidence of rejection? Neonatal status: Percentile of Birthweight, Apgar & Congenital malformations.

Author:  In Case 1, placenta preavia marginalis was present. This was added to the text: Line 337. Otherwise there was no problem with placentation – we did not mention this explicitly in the text.

Placenta sampling was not performed nor mentioned in the text.

Neonatal status: We added this info: “There were no congenital malformations nor neonatal complications” Line 321:

Percentile of Birthweight and Apgar: this info was added to each of the babies: lines 335, 346 and 359. Our national birth weight chart give a percentile interval for boys and girls, this interval value was presented.  

  1. Page 10, Table 303. What is exactly the pathophysiologic association between embryo-transfer and renal impairment. Please provide reference.

Author: Yes, there is probably no association between ET a CKD. The reason for renal impairment was the toxic effect of immune suppression as stated in Line 306-307.

The sentence was adapted as follows: “Owing to numerous unsuccessful ETs, as well as gaps between the transfers, resulting in prolonged exposure to immune suppression, the recipient developed renal impairment.”  Line 333

Additional changes to the manuscript

Typographical errors, e.g.:

Line 157: “Two venous outflow…” replaced by: “Venous outflow…“

„In case 4…“ was replaced by: „In Case 4…“ This capital letter (Case) was replaced throughout the whole manuscript in descriptions of a particular case, e.g. Case 4.

New sentence, line 446-448 “The first UTx from a DD ever had used a nulliparous DD and this was also the second UTx case worldwide, carried out in Turkey in 2011. This case produced clinical pregnancy but ended in a miscarriage.”

Line 390-92: new sentence: “There were no rejection episodes during pregnancies in our study. Three cases that resolved by steroid therapy have been reported worldwide, namely in Gothenburg, Sweden [1], Cleveland, Ohio [12] and Dallas, Texas [26].”

Round 2

Reviewer 1 Report

The authors responded sufficiently to all of my questions and comments. Many thanks. The quality of the manuscript significantly improved after revision. I have no further comments.